# Resident and recruited macrophages differentially contribute to cardiac healing after myocardial ischemia

Tobias Weinberger[1,2,3,4], Messerer Denise[1,2], Markus Joppich[5], Maximilian Fischer[1,2,3], Clarisabel Garcia Rodriguez[4], Konda Kumaraswami[1,2], Vanessa Wimmler[1,2], Sonja Ablinger[1,2], Saskia Räuber[1,2,6], Jiahui Fang[1,2], Lulu Liu[1,2], Wing Han Liu[1], Julia Winterhalter[1,2], Johannes Lichti[1], Lukas Thomas[1,2,3], Dena Esfandyari[3,7], Guelce Percin[8], Sandra Matin[9], Andrés Hidalgo[9,10], Claudia Waskow[8,11], Stefan Engelhardt[3,7], Andrei Todica[12], Ralf Zimmer[5], Clare Pridans[13,14], Elisa Gomez Perdiguero[4], Christian Schulz[1,2,3,15]*

[1]Medical Clinic I., Department of Cardiology, University Hospital, Ludwig Maximilian University, Munich, Germany; [2]Institute of Surgical Research at the Walter-Brendel-Centre of Experimental Medicine University, Munich, Germany; [3]DZHK (German Centre for Cardiovascular Research), Partner site Munich Heart Alliance, Munich, Germany; [4]Institut Pasteur, Unité Macrophages et Développement de l'Immunité, Département de Biologie du Développement et Cellules Souches, Paris, France; [5]LFE Bioinformatik, Department of Informatics, Ludwig Maximilian University, Munich, Germany; [6]Department of Neurology, Medical Faculty, Heinrich Heine University of Düsseldorf, Düsseldorf, Germany; [7]Institute of Pharmacology and Toxicology, Technical University Munich, Munich, Germany; [8]Immunology of Aging, Leibniz-Institute on Aging - Fritz-Lipmann-Institute (FLI), Jena, Germany; [9]Area of Cell & Developmental Biology, Centro Nacional de Investigaciones Cardiovasculares Carlos III, Madrid, Spain; [10]Vascular Biology and Therapeutics Program and Department of Immunobiology, Yale University School of Medicine, New Haven, United States; [11]Faculty of Biological Sciences, Friedrich-Schiller-University, Jena, Germany; [12]Department of Nuclear Medicine, Ludwig Maximilian University, Munich, Germany; [13]Simons Initiative for the Developing Brain, Centre for Discovery Brain Sciences, University of Edinburgh, Edinburgh, United Kingdom; [14]University of Edinburgh Centre for Inflammation Research, The Queen's Medical Research Institute, Edinburgh, United Kingdom; [15]Department of Immunopharmacology, Mannheim Institute for Innate Immunoscience (MI3), Medical Faculty Mannheim, Heidelberg University, Mannheim, Germany

*For correspondence: Christian.Schulz@medma.uni-heidelberg.de

**Abstract** Cardiac macrophages are heterogenous in phenotype and functions, which has been associated with differences in their ontogeny. Despite extensive research, our understanding of the precise role of different subsets of macrophages in ischemia/reperfusion (I/R) injury remains incomplete. We here investigated macrophage lineages and ablated tissue macrophages in homeostasis and after I/R injury in a CSF1R-dependent manner. Genomic deletion of a fms-intronic regulatory element (FIRE) in the *Csf1r* locus resulted in specific absence of resident homeostatic and antigen-presenting macrophages, without affecting the recruitment of monocyte-derived macrophages to the infarcted heart. Specific absence of homeostatic, monocyte-independent macrophages altered the immune cell crosstalk in response to injury and induced proinflammatory neutrophil polarization,

resulting in impaired cardiac remodeling without influencing infarct size. In contrast, continuous CSF1R inhibition led to depletion of both resident and recruited macrophage populations. This augmented adverse remodeling after I/R and led to an increased infarct size and deterioration of cardiac function. In summary, resident macrophages orchestrate inflammatory responses improving cardiac remodeling, while recruited macrophages determine infarct size after I/R injury. These findings attribute distinct beneficial effects to different macrophage populations in the context of myocardial infarction.

## eLife assessment

Using state-of-the-art fate-mapping models and genetic and pharmacological targeting approaches, this study provides **important** findings on the distinct functions exerted by resident and recruited macrophages during cardiac healing after myocardial ischemia. Evidence supporting the conclusions are **solid** with the use of the FIRE mouse model in combination with fate-mapping to target fetal-derived macrophages. This study will be of interest for the macrophage biologists working in the heart but also in others tissues in the context of inflammation.

## Introduction

Macrophages are important effectors of innate immunity. They are essential for host defense against infections but are also involved in different cardiovascular diseases. They represent the most abundant immune cell population in healthy cardiovascular tissues (*Heidt et al., 2014*; *Weinberger et al., 2020*), where they contribute to organ functions (*Hulsmans et al., 2017*) and maintenance of tissue homeostasis (*Nicolás-Ávila et al., 2020*). In cardiovascular diseases such as atherosclerosis and its main sequelae, ischemic stroke and acute myocardial infarction (AMI), macrophage functions are central to both disease development and healing. AMI has remained a leading cause of mortality and morbidity worldwide (*Ahmad and Anderson, 2021*; *Lozano et al., 2012*). Although acute survival in this condition has improved through the broad availability of percutaneous coronary intervention, adverse myocardial remodeling, and fibrosis frequently result in heart failure (*Gerber et al., 2016*). Pathophysiologically, the diminished blood supply to myocardial tissue during AMI leads to acute tissue necrosis, which induces a profound sterile inflammation and triggers complex cascade of immune processes and tissue remodeling (*Hilgendorf et al., 2014*; *Honold and Nahrendorf, 2018*; *Nahrendorf et al., 2007*). Consequently, uncontrolled immune reactions in the course of AMI are associated with impaired wound healing and adverse remodeling and can result in worsened cardiac outcome (*Panizzi et al., 2010*).

Macrophages play an essential role in cardiac injury, and thus represent a potential therapeutic target (*Hilgendorf et al., 2014*; *Nahrendorf et al., 2007*). However, they are an heterogenous population (*Kubota et al., 2019*; *Weinberger and Schulz, 2015*; *Zaman and Epelman, 2022*), and a large body of work has shown that they can have both pro- and anti-inflammatory functions. The differential roles of macrophage populations in AMI have remained incompletely understood. Cardiac macrophages can derive from embryonic and adult hematopoietic progenitors (*Epelman et al., 2014*). Fate-mapping analyses have identified yolk sac (YS) erythro-myeloid progenitors (EMPs) as a principal source of cardiac macrophages in adult life (*Ginhoux et al., 2010*). However, limited labeling in inducible cre reporter systems has not allowed for precisely differentiating and quantifying developmental origins of cardiac macrophages. Furthermore, targeting of these macrophages has been challenging (*Frieler et al., 2015*; *Ruedl and Jung, 2018*).

In this study, we investigated the cellular identity of cardiac macrophages in association with their developmental paths and their immune responses to ischemia/reperfusion (I/R) injury. By combining lineage tracing with single-cell RNA sequencing, we provide an in-depth analysis of the differential functions of resident and recruited cardiac macrophages. We then harnessed mice with genomic deletion of the fms-intronic regulatory element (FIRE) (*Rojo et al., 2019*), that allowed us to specifically address populations of resident macrophages in the infarcted heart and compared them to mice, in which both recruited and resident macrophages are depleted by pharmacological inhibtion of the CSF1R-signaling pathway. Using these approaches of selective

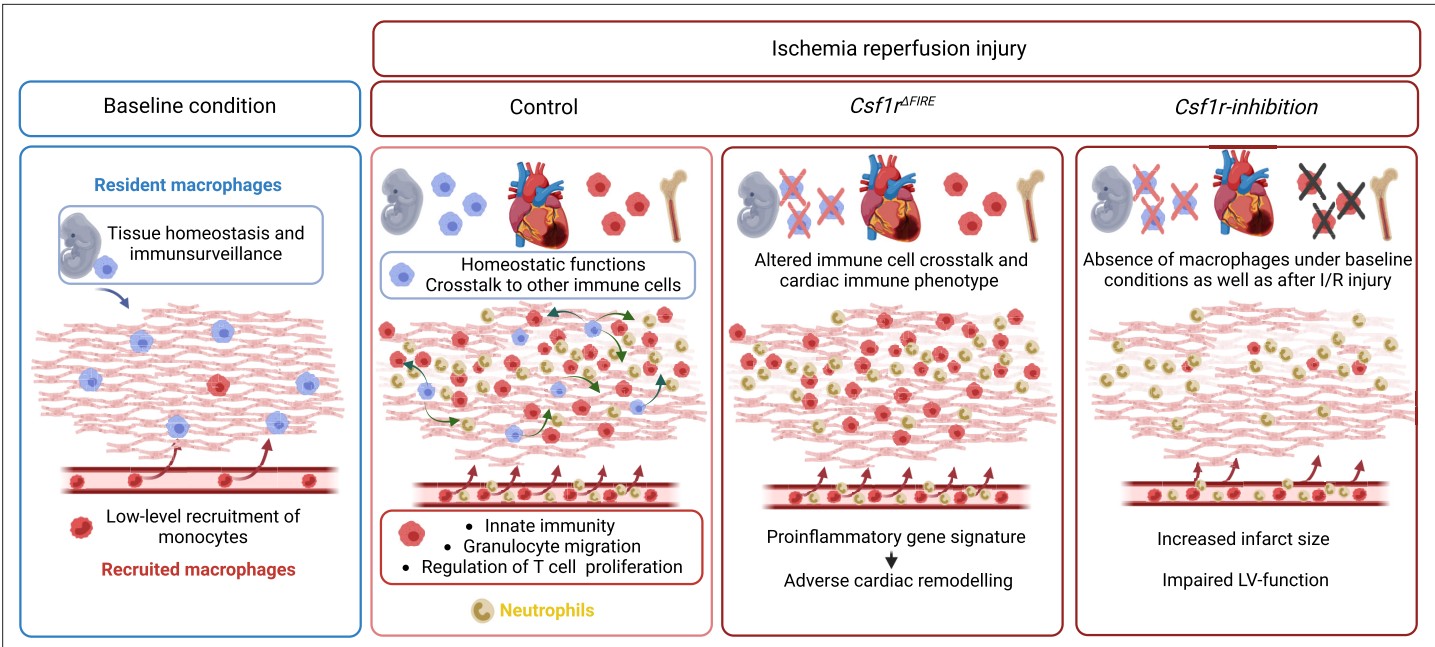

**Figure 1.** Graphical abstract.

macrophage depletion, we could attribute different beneficial functions to resident and also to recruited macrophages which impact differently on cardiac remodeling, infarct size, and cardiac outcome (*Figure 1*).

## Results
### Absence of resident cardiac macrophages in *Csf1r*^ΔFIRE^ mice

To quantify the contribution of YS EMPs to cardiac resident macrophages, we harnessed constitutive labeling in *Tnfsf11a^Cre^;Rosa26^fs-eYFP^* mice (*Jacome-Galarza et al., 2019*; *Mass et al., 2016*; *Percin et al., 2018*). ~80% of cardiac macrophages expressed YFP in hearts of 12-week-old animals, whereas blood monocytes were not labeled (*Figure 2A*; *Figure 2—figure supplement 1*). EMP-derived microglia served as control confirming high efficiency of Cre-mediated recombination. In a comparative analysis, we traced fetal and adult definitive hematopoiesis in *Flt3^Cre^Rosa26^fs-eYFP^* mice (*Gomez Perdiguero et al., 2015*; *Schulz et al., 2012*), indicating that ~20% of cardiac macrophages derived from definitive HSC (*Figure 2B*). Thus, the majority of resident macrophages in the healthy heart is of early embryonic origin.

To determine the role of resident macrophages in the mouse heart, we generated *Tnfsf11a^Cre^Rosa26^fs-DTR^* mice that express the avian diphtheria toxin (DT) receptor in the EMP lineage, rendering them susceptible to DT-mediated ablation. Administration of a single DT dose, however, resulted in premature death of all mice (*n* = 6) within 24 hr (*Figure 2—figure supplement 2*). The injected mice presented with a systemic inflammatory response syndrome-like phenotype, indicating that acute depletion of EMP-derived macrophages is not viable.

In *Csf1r*^ΔFIRE^ mice, genetic deletion of the FIRE in the first intron of the *Csf1r* gene leads to a selective absence of tissue macrophages without the developmental defects observed in *Csf1r*^−/−^ mice (*Munro et al., 2020*; *Rojo et al., 2019*). We first confirmed that cardiac macrophages in *Csf1r*^ΔFIRE/ΔFIRE^ mice (further termed ΔFIRE) mice were reduced by ~90% and the number of monocytes and neutrophils was not altered (*Figure 2C–G*). We thus crossed the ΔFIRE animals with the YS EMP fate-mapping line (*Tnfsf11a^Cre^Rosa26^fs-eYFP^* mice) and showed that the macrophages absent in ΔFIRE animals were EMP-derived macrophages (*Figure 2E*), thus offering a new model to investigate the role of resident macrophages in cardiac injury.

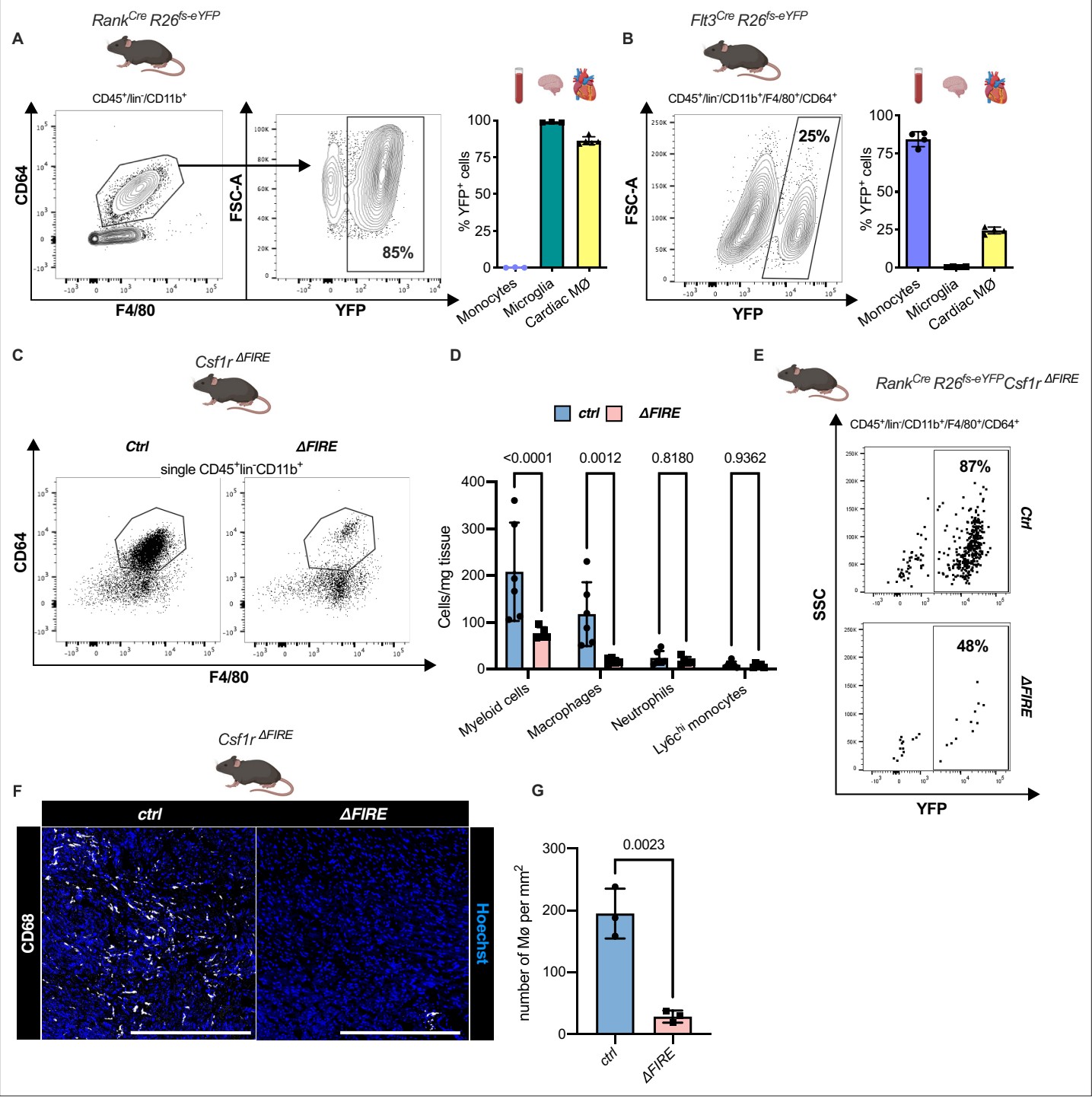

**Figure 2.** Absence of resident cardiac macrophages in *Csf1r^ΔFIRE^* mice. (**A**) Flow cytometry analysis of 3-month-old *Tnfsf11a^Cre^Rosa26^fs-eYFP^* mice, showing single/CD45⁺/lin-(CD11c, Ter119, Tcrß, Nk1.1)/CD11b⁺ cardiac cells, eYFP expression in macrophages (CD64⁺/F4/80⁺) and percentage of eYFP⁺ blood monocytes, microglia and cardiac macrophages (*n* = 3–5 each from an independent experiment). (**B**) Flow cytometry analysis of 3-month-old *Flt3^Cre^Rosa26^fs-eYFP^* mice, showing macrophage expression of eYFP and percentage of eYFP⁺ blood monocytes, microglia, and cardiac macrophages (*n* = 4 each from an independent experiment). (**C**) Representative flow cytometry analysis of cardiac macrophages in *control* and *ΔFIRE mice*. (**D**) Quantification of myeloid cells by flow cytometry (CD45⁺/lin⁻/CD11b⁺), macrophages (CD45⁺/lin⁻/CD11b⁺/CD64⁺/F4/80⁺), neutrophils (CD45⁺/lin⁻/CD11b⁺/CD64⁻/F4/80⁻/Ly6g⁺), and Ly6c^hi^ monocytes (CD45⁺/lin⁻/CD11b⁺/CD64⁻/F4/80⁻/Ly6g⁻/Ly6c^hi^) (*n* = 6 for control and *n* = 5 for *ΔFIRE* mice, each single experiments). (**E**) Representative flow cytometry analysis of cardiac macrophages and their expression of eYFP in *Csf1r^ΔFIRE/+^Tnfsf11a^Cre^Rosa26^fs-eYFP^* and *Csf1r^ΔFIRE/ΔFIRE^Tnfsf11a^Cre^Rosa26^fs-eYFP^*. (**F**) Representative immunohistological images showing macrophages (CD68⁺ cells in white and Hoechst in blue)

*Figure 2 continued on next page*

*Figure 2 continued*

in *control* and *ΔFIRE* hearts in 3-month-old mice at baseline conditions (scale bars represent 500 μm). (**G**) Quantification of macrophages by histology (n = 3 for control and *ΔFIRE* mice). Either Fisher's Least Significant Difference (LSD) test or unpaired *t*-test was performed and mean ± standard deviation (SD) is shown.

The online version of this article includes the following figure supplement(s) for figure 2:

**Figure supplement 1.** Gating strategy for cardiac myeloid immune cells.

**Figure supplement 2.** Diphtheria toxin (DT)-mediated depletion of erythro-myeloid progenitor (EMP)-derived macrophages in *Tnfsf11a^CreRosa26^fs-DTR* mice.

## Changes in the cardiac immune phenotype in Csf1r^ΔFIRE mice in baseline conditions

To further characterize the macrophage populations absent in healthy hearts of adult ΔFIRE mice, we carried out single-cell RNA-sequencing (scRNA-seq) of CD45+ immune cells in wildtype, ΔFIRE, and *Tnfsf11a^CreRosa26^fs-eYFP* mice (*Figure 3A*, *Figure 3—figure supplement 1*). We identified the presence of six macrophage clusters in control mice (*Figure 3B*, *Figure 3—figure supplement 1*). Two clusters of macrophages expressing homeostatic (e.g. *Lyve1*, *Gas6*, *Stab1*) and antigen presentation-related genes (e.g. *Cd74*, *H2-Ab1*) were specifically ablated in ΔFIRE mice (*Figure 3C*), which largely represent YFP-expressing macrophages in *Tnfsf11a^CreRosa26^fs-eYFP* mice (*Figure 3D*). Specifically, we mapped ~90% of *yfp* transcripts to the two clusters of homeostatic and antigen-presenting macrophages that were ablated in ΔFIRE. Other clusters were not reduced in ΔFIRE, and also were not quantitatively represented in the YS macrophage lineage tracing model, that is Cx3cr1^hi (2% YFP), Ccr2^lowLy6c^lo (5% YFP), and Ccr2^hiLy6c^hi (1% YFP) macrophages within respective clusters (*Figure 3D*). The specific impact of ΔFIRE on resident macrophages is further supported by the observation, that no changes in gene expression nor cell numbers were observed in Ly6c^hi monocytes and inflammatory Ccr2^hiLy6c^hi macrophages (*Figure 3C*, *Figure 3—figure supplement 2*). Together, ΔFIRE allows for investigating the role of resident cardiac macrophages that are largely of YS origin.

Absence of resident macrophages in ΔFIRE was associated with changes in gene expression in non-macrophage clusters such as *T- and B-cells* and *natural killer (NK)* cells (*Figure 3—figure supplement 3*). Furthermore, overall phagocytic capacity, as inferred by expression of phagocytosis-related genes (*Amorim et al., 2022*), was reduced (*Figure 3E*). This suggests that absence of resident macrophages is accompanied by distinct changes in immune functions in the homeostatic heart.

## Adverse cardiac remodeling in *Csf1r^ΔFIRE* mice after I/R injury

To assess the impact of the absence of resident macrophages in cardiac injury, we subjected ΔFIRE mice to I/R injury and investigated remodeling and functional outcome by sequential positron emission tomography (PET) imaging after 6 and 30 days (*Figure 4A*). Function of the cardiac left ventricle (LV), as determined by ejection fraction (LVEF) and stroke volume (SV), improved in controls in the course of post-I/R remodeling (*Figure 4B, C*). In contrast, LVEF and SV remained unchanged or worsened in ΔFIRE mice, and longitudinal observations of mice indicated a negative net effect on ejection fraction from day 6 to 30 post I/R. Thus, absence of resident macrophages negatively influenced cardiac remodeling in the course of infarct healing (*Figure 4D–G*). Infarct size as determined by viability defect (PET) and fibrotic area (histology) was not different after 30 days (*Figure 4E–H*).

## Recruitment of BM-derived macrophages into infarct zone of *Csf1r^ΔFIRE* mice

To gain a deeper understanding of the inflammatory processes taking place in the infarcted heart, we quantified macrophage distribution by immunofluorescence and flow cytometry analysis of ischemic and remote areas after I/R. In ΔFIRE mice, macrophages were largely absent in the remote zone of infarcted hearts (*Figure 5A*), indicating sustained depletion of resident macrophages. However, macrophages strongly increased in the infarct area and their numbers were not different in both infarct and border zones between ΔFIRE and control mice (*Figure 5A*). This indicated recruitment and differentiation of ΔFIRE-independent macrophages from the circulation into these regions. Indeed, complementing lineage tracing of bone marrow (BM) hematopoietic stem cells (HSC) in *Flt3^Cre* mice (*Figure 5B*) and YS EMP in *Tnfsf11a^Cre* mice (*Figure 5C*) proved the recruitment of macrophages from

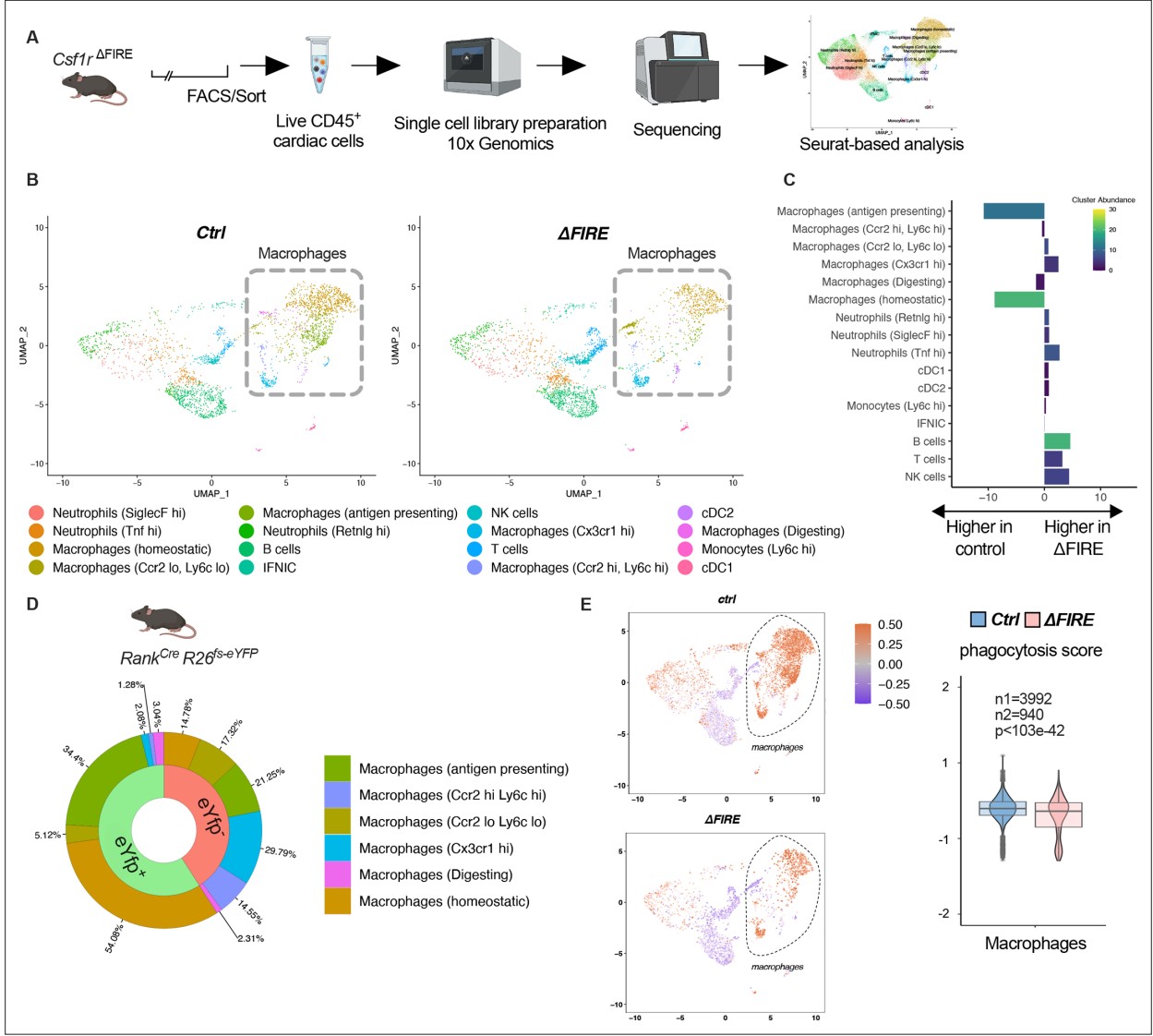

**Figure 3.** Changes in the cardiac immune phenotype in *Csf1r*^*ΔFIRE*^ mice in baseline conditions. (**A**) Experimental setup to analyze cardiac immune cells using scRNA-seq of sorted CD45⁺/live cells. (**B**) UMAPs (Uniform Manifold Approximation and Projection) of control and ΔFIRE in baseline conditions (*n* = 3 for *control* and *ΔFIRE*). (**C**) Absolute difference (percentage points) in cluster abundance between *control* and *ΔFIRE*. (**D**) Contribution of erythromyeloid progenitor (EMP)-derived (*eYfp* expressing) macrophages to the different macrophage clusters analyzed by scRNA-seq analysis of immune cells harvested from a *Tnfsf11a*^*Cre*^*Rosa26*^*fs-eYFP*^ mouse. (**E**) Phagocytosis score projected on a UMAP displaying *control and ΔFIRE* immune cell subsets. Violin and box plots show the computed phagocytosis score combined in all macrophage clusters (*n1/n2* represents number of cells from control/*ΔFIRE* mice).

The online version of this article includes the following figure supplement(s) for figure 3:

**Figure supplement 1.** Clustering of cardiac immune cells in single-cell RNA analysis.

**Figure supplement 2.** Differential gene expression in monocyte and macrophage clusters in baseline conditions in *control* and *ΔFIRE* mice.

**Figure supplement 3.** Differential gene expression in non-macrophage clusters in baseline conditions in *control* and *ΔFIRE* mice.

BM HSC (Flt3 GFP+ and Tnfsf11a RFP−, respectively) in the early phase of I/R injury. 30 days after I/R, BM-derived macrophages remained overrepresented in the infarct zone (~75% HSC contribution), and differential contribution of BM HSC declined from the border zone (~50%) to the remote zone (~35%) (*Figure 5D*). Taken together, recruited BM-derived macrophages represent the main population in the infarct area and their recruitment was unaltered in ΔFIRE mice. This supports the notion that resident macrophages influence cardiac remodeling but recruited macrophages drive infarct size after I/R.

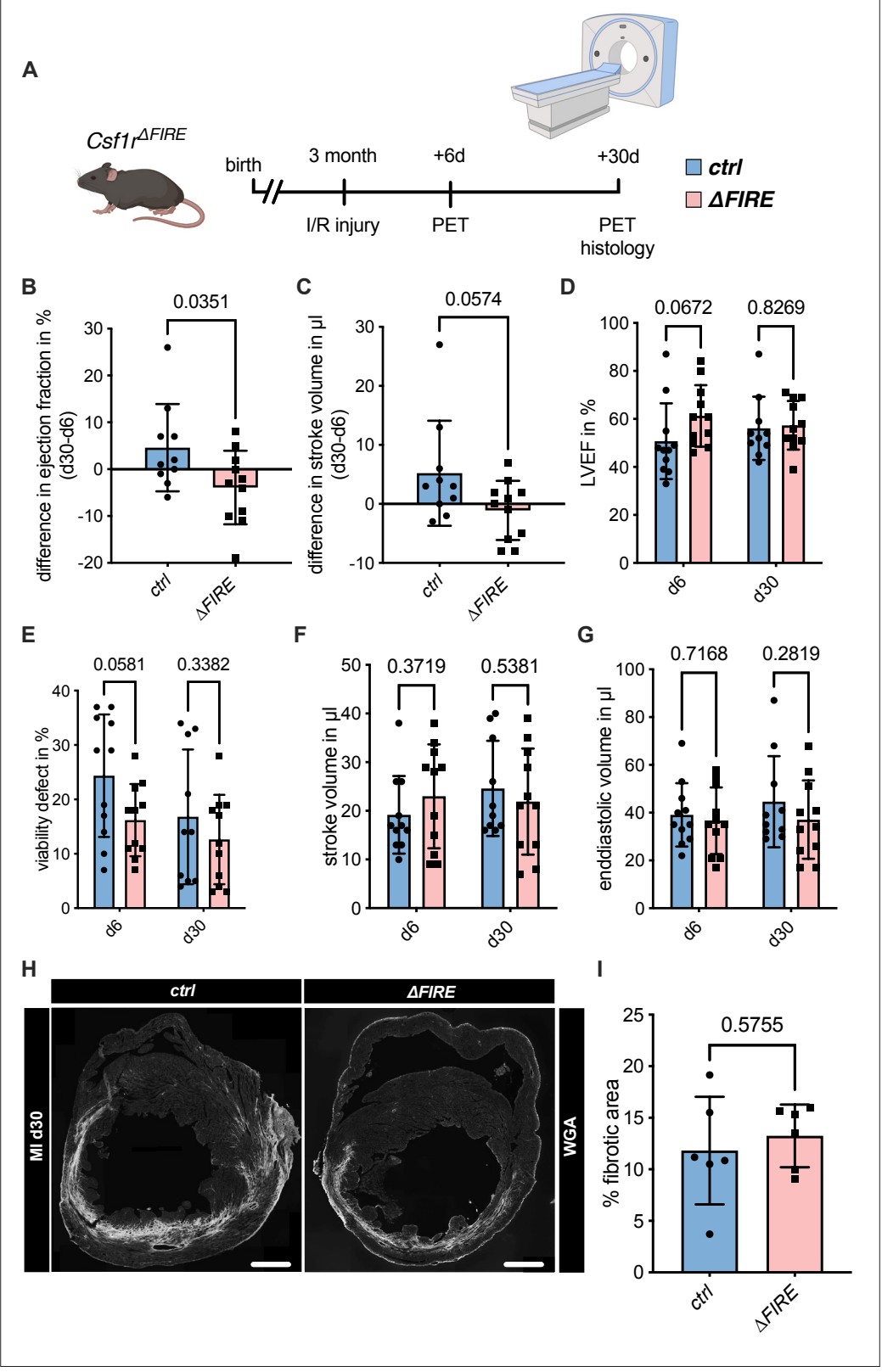

**Figure 4.** Adverse cardiac remodeling in *Csf1r^{ΔFIRE}* mice after ischemia/reperfusion (I/R) injury. (**A**) Schematic of the sequential analysis of cardiac function, dimensions, and viability using positron emission tomography 6 and 30 days after I/R injury in *control* and *ΔFIRE* with (**B, C**) showing the intraindividual changes in each parameter from d6 to d30 and (**D–G**) the individual time points on d6 and d30 (d6: *n* = 11 for *control and ΔFIRE*, d30: *n* = 10

*Figure 4 continued on next page*

*Figure 4 continued*

for *control* and *n* = 11 *for ΔFIRE*). (**B, D**) left ventricular ejection fraction (LVEF), (**C, E**) percentage of the viability defect, (**F**) stroke volume, and (**G**) left ventricular end-diastolic volume. (**H**) Representative immunohistological images showing the fibrotic area (WGA⁺ area) in hearts from *control and ΔFIRE* mice 30 days after I/R injury. Right panel shows the percentage of fibrotic area in the respective groups (*n* = 6 for each group). Student's *t*-test was performed and mean ± standard deviation (SD) is shown.

## Transcriptional landscape of resident versus recruited macrophages in I/R injury

To address the differential responses of resident and recruited macrophages to I/R injury, we generated BM chimeric mice. We applied an irradiation-independent model using conditional deletion of c-myb to deplete BM hematopoietic cells in CD45.2 mice and replace them with CD45.1 donor HSC (hematopoietic stem cells). Two days after I/R injury, we FACS-sorted recruited (CD45.1⁺) and resident (CD45.2⁺) macrophages and carried out bulk RNA sequencing (**Figure 6A**). The two macrophage populations exhibited profound transcriptional differences after I/R injury. Expression of homeostasis-related genes like *Timd4*, *Lyve1*, *Cd163*, and *Retnla* was increased in resident CD45.2⁺ macrophages (**Figure 6B, C**). Substantiating the differential regulation of macrophage programs, CD45.1⁺ macrophages increased inflammatory- and host defense-related gene ontology (GO) terms (e.g. myeloid leukocyte-related immunity, killing of cells), whereas CD45.2⁺ macrophages upregulated development- and homeostasis-related GO terms (e.g. extracellular matrix organization, vasculature development, heart muscle development) (**Figure 6D**). GO enrichment analysis further predicted that processes involved in innate (e.g. leukocyte migration, regulation of defense response, response to bacterium) as well as the adaptive immunity (e.g. T cell selection, T cell activation, B cell proliferation) were increased in CD45.1⁺ macrophages after infarction (**Figure 6D**). Thus, resident and recruited macrophages provide distinct transcriptional changes and inferred functions in response to I/R.

## Altered inflammatory patterns and immune cell communication in *Csf1r*^ΔFIRE mice

To evaluate the immune response of resident and recruited macrophages to I/R injury in ΔFIRE mice, we interrogated the transcription profile of CD45⁺ cells from the infarct area (**Figure 7A**). In contrast to the absence of homeostatic and antigen-presenting macrophage clusters in healthy hearts of ΔFIRE mice (**Figure 3B, C**), there were less differences in immune cell clusters between ΔFIRE and control mice 2 days after I/R (**Figure 7**). Abundance of homeostatic macrophages was also reduced at this time point, however, other clusters including *Ccr2*^hi*Ly6c*^hi inflammatory macrophages were not altered, which is in line with our histological findings.

ΔFIRE was associated with some changes in gene expression in cardiac non-macrophage immune cells. Across different clusters, including lymphocyte and neutrophil clusters, expression of anti-inflammatory genes like *Chil3 (Ym1)* and *Lcn2* was reduced. Vice versa, expression of Bcl-family genes like *Blc2a1a* and *Bcl2a1d*, which are associated with apoptosis and inflammatory pathways, was higher in ΔFIRE mice (**Figure 7—figure supplements 1–3**). Other upregulated genes were related to antigen presentation (e.g. *Cd74*, *H2-Ab1*), as identified in the *Ccr2*^lo*Ly6c*^lo, homeostatic and *Ccr2*^hi*Ly6c*^hi macrophage clusters (**Figure 7—figure supplements 1 and 2**). Furthermore, we interrogated inflammatory gene expression in neutrophils, which are abundant first responders to myocardial infarction. We found that a computed score summing up inflammasome activation (**Amorim et al., 2022**) was increased in all neutrophil clusters in ΔFIRE mice (**Figure 7D**). Thus, the absence of cardiac macrophages was associated with altered inflammatory properties of non-macrophage immune cells in the infarcted heart.

Altered intercellular crosstalk of macrophages is a hallmark of cardiac inflammation. We therefore assessed ligand–receptor (LR) interactions between immune cell populations after I/R injury. Indeed, the number of LR interactions with neutrophils and lymphocytes, as well as the strength of the macrophage-emitted communication signals was markedly reduced in homeostatic, antigen-presenting, and *Ccr2*^lo*Ly6c*^lo macrophage clusters (**Figure 7E, F**, **Figure 7—figure supplement 4**). In contrast, immune cell communication in BM-derived macrophage clusters (e.g. *Ccr2*^hi*Ly6c*^hi macrophage cluster, **Figure 7—figure supplement 5**) was not different to control mice. Taken together,

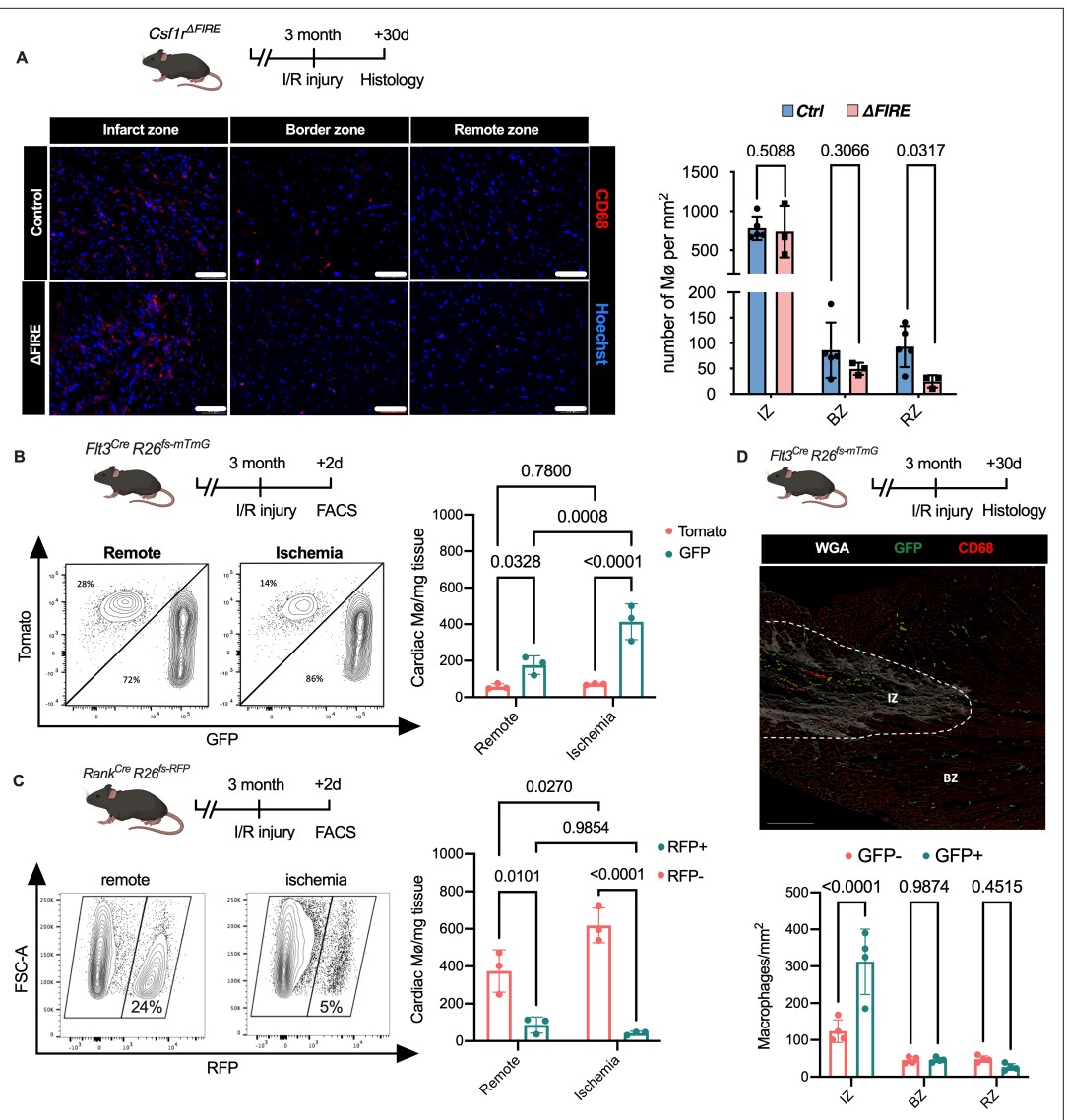

**Figure 5.** Recruitment of BM-derived macrophages into infarct zone of *Csf1r^ΔFIRE^* mice. (**A**) Representative immunohistology of hearts from *ΔFIRE* mice 30 days after ischemia/reperfusion (I/R) injury showing macrophages (CD68⁺ cells in red and Hoechst in blue) in the infarct, border and remote zone. Right panel shows number of cardiac macrophages in the respective area (*n* = 5 *control* and *n* = 3 for *ΔFIRE*). (**B**) Flow cytometry analysis of *Flt3^Cre^Rosa26^fs-mT/mG^* mice 2 days after I/R injury, (left) representative flow cytometry showing expression of tomato and GFP in macrophages in the remote and ischemic myocardium and (right) number of tomato⁺ and GFP⁺ cardiac macrophages in the respective area (*n* = 3, each individual experiment). (**C**) Flow cytometry analysis of *Tnfsf11a^Cre^Rosa26^RFP^* mice 2 days after I/R injury, (left) representative flow cytometry showing expression of RFP in macrophages in the remote and ischemic myocardium and (right) number of RFP⁻ and RFP⁺ cardiac macrophages in the respective area (*n* = 3, each individual experiment). (**D**) Histological analysis of *Flt3^Cre^Rosa26^fs-mT/mG^* mice 30 days after I/R injury in the infarct, border, and remote zones, (left) representative immunohistology of the infarct and border zones and (right) number of GFP⁻ and GFP⁺ cardiac macrophages in the respective areas (*n* = 4). Fisher's Fisher's Least Significant Difference (LSD) test was performed for all experiments and mean ± standard deviation (SD) is shown.

deficiency in resident macrophages in I/R injury altered the intercellular immune crosstalk and induced a proinflammatory signature in for example cardiac neutrophils. However, transcriptional profiles of BM-derived inflammatory macrophages were largely unaltered. Together with our histological analysis showing the dominance of recruited BM-derived macrophages in the early phase of I/R, this potentially explained the limited impact of ΔFIRE on functional outcome after I/R injury.

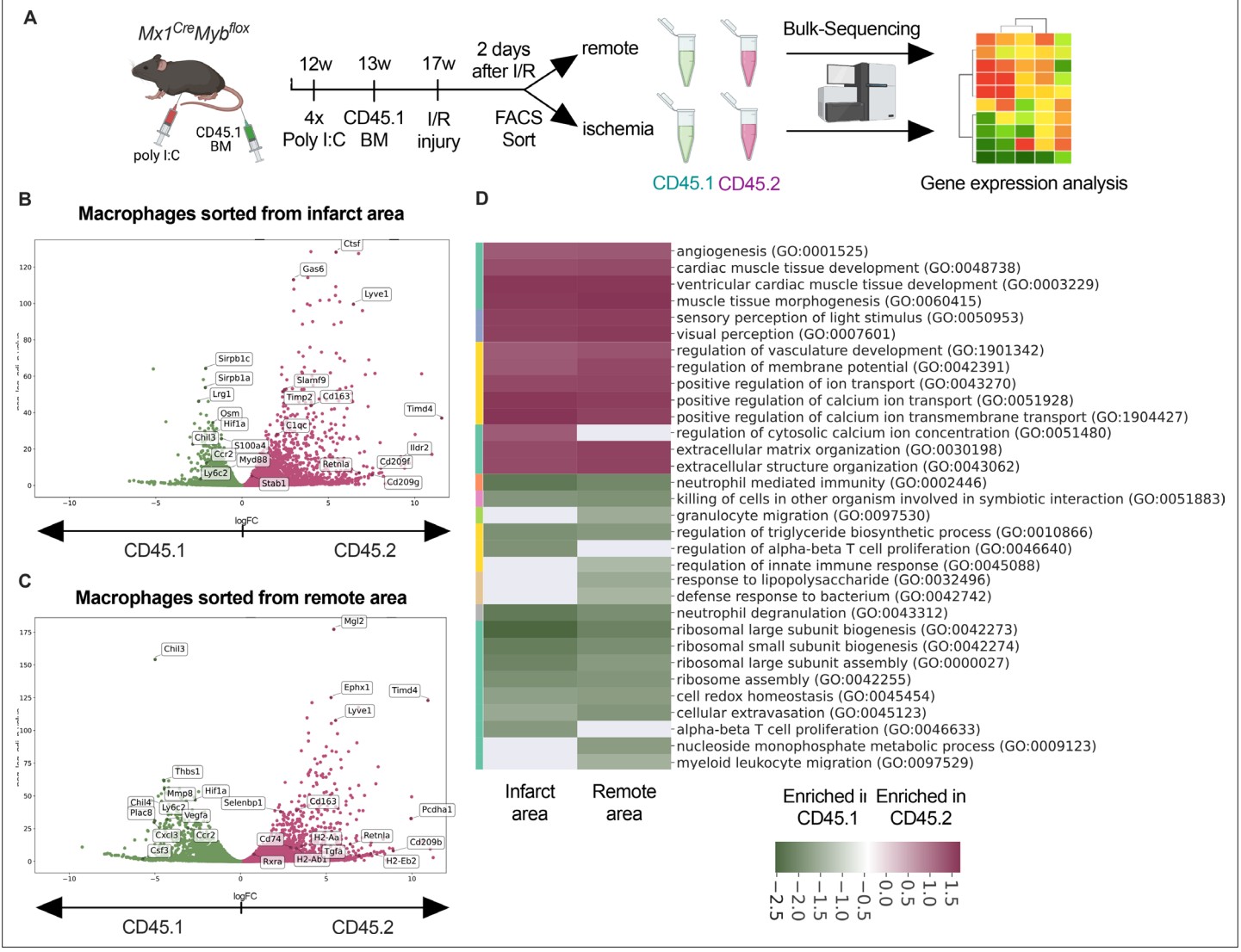

**Figure 6.** Transcriptional landscape of resident versus recruited macrophages in ischemia/reperfusion (I/R) injury. (**A**) Experimental setup to generate non-irradiation BM chimera using *CD45.2 Mx1^Cre^Myb^flox/flox^* and transplantation of CD45.1 BM. I/R injury was induced 4 weeks after BM transplantation and CD45.1⁺ and CD45.2⁺ macrophages were sorted from the remote and ischemic myocardium 2 days after I/R injury and RNA-sequencing was performed on bulk cells (n = 3). Volcano plot showing differential gene expression analysis results of recruited CD45.1 versus resident CD45.2 macrophages in the (**B**) remote and (**C**) ischemic zones. (**D**) Gene ontology enrichment analysis showing specific biological processes enriched in CD45.1 and CD45.2 macrophages in the ischemic and remote zones.

## Ablation of resident and recruited macrophages severely impacts on cardiac healing after I/R injury

To test this hypothesis, we determined the effect of combined ablation of resident and recruited macrophages. We therefore exposed mice to continuous treatment with the CSF1R-inhibitor PLX5622 (*Figure 8A*). In healthy hearts, inhibitor treatment resulted in the absence of cardiac macrophages within 7 days (*Figure 8—figure supplement 1*). We then subjected mice to I/R injury and investigated outcome by sequential PET imaging and histology (*Figure 8A*). Treatment with PLX5622 diminished macrophage numbers in both remote and infarct areas in the early phase (day 2) after injury. Recruitment of other myeloid cells for example neutrophils was not altered in this context (*Figure 8B*). This effect was pronounced in the chronic phase (day 30) after I/R injury, in which macrophages were largely absent in remote, border, and infarct zones (*Figure 8C, D*). Absence of resident and recruited macrophages was associated increased infarct size, as determined by fibrosis area (WGA histology) as

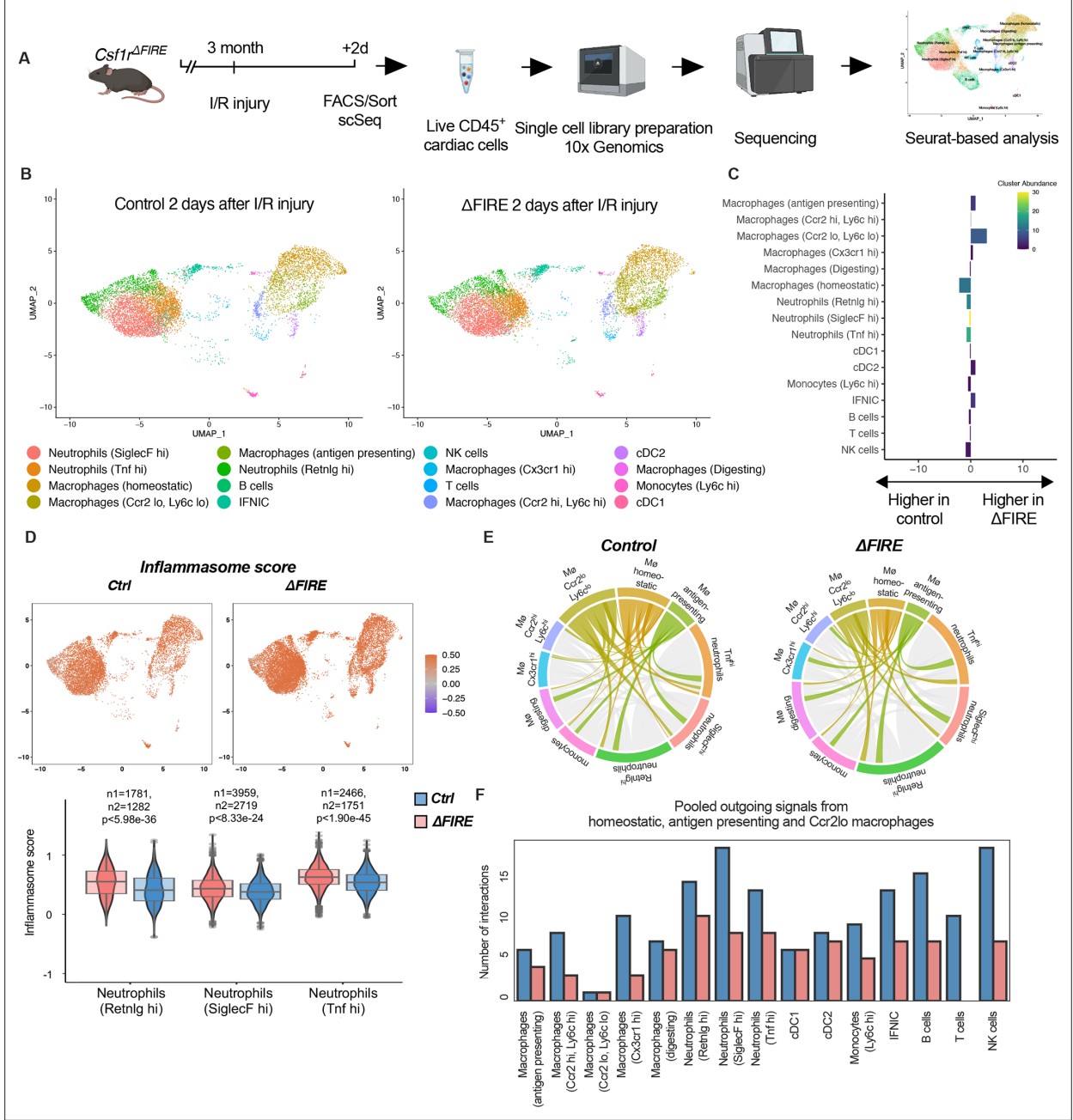

**Figure 7.** Altered inflammatory patterns and immune cell communication in *Csf1r^ΔFIRE* mice. (**A**) Experimental setup to analyze transcriptional changes in cardiac immune cells on a single-cell level 2 days after ischemia/reperfusion (I/R) injury in *ΔFIRE* mice. (**B**) UMAPs of control and ΔFIRE 2 days after I/R injury (*n* = 2 for *control* and *ΔFIRE*). (**C**) Absolute difference (percentage points) in cluster abundance between *control* and *ΔFIRE*. (**D**) Inflammasome score projected on a UMAP displaying control and ΔFIRE immune cell subsets after I/R injury. Violin and box plots show the computed inflammasome score in neutrophil clusters (n1/n2 represents number of cells from control/ *ΔFIRE* mice). (**E**) Ligand–receptor interactions of *antigen-presenting, Ccr2^lo ly6c^lo and homeostatic macrophages* (highlighted) with other immune cell clusters. Shown are the aggregated communication scores (width of interactions) for all cell types. Only communication scores larger than 6 are considered. (**F**) Number of interactions (with communication score >6) outgoing from homeostatic, antigen-presenting and Ccr2^lo macrophages to other immune cell clusters.

The online version of this article includes the following figure supplement(s) for figure 7:

**Figure supplement 1.** Violin plots comparing expression of Lcn2, Chil3, Cd74, and Bcl2a1a in the different immune cell clusters after ischemia/ reperfusion (I/R) in *control* and *ΔFIRE* mice.

**Figure supplement 2.** Differential gene expression in monocyte and macrophage clusters after ischemia/reperfusion (I/R) in *control* and *ΔFIRE* mice.

**Figure supplement 3.** Differential gene expression in non-macrophage clusters after ischemia/reperfusion (I/R) in in *control* and *ΔFIRE* mice.

*Figure 7 continued on next page*

*Figure 7 continued*

**Figure supplement 4.** Different outgoing signals in macrophage subpopulations in ischemia/reperfusion (I/R) injury in *Csf1r^ΔFIRE/+* and *Csf1r^ΔFIRE/ΔFIRE* mice.

**Figure supplement 5.** Similar outgoing signals in macrophage subpopulations in ischemia/reperfusion (I/R) injury in *Csf1r^ΔFIRE/+* and *Csf1r^ΔFIRE/ΔFIRE* mice.

well as viability defect (PET), and resulted in deterioration of cardiac function (*Figure 8E–I*). Specifically, LVEF was reduced 6 days after I/R, and remained strongly impaired at 30 days (*Figure 8E*).

Taken together, sole absence of resident macrophages had limited negative impact on cardiac remodeling. Absence of both resident and recruited macrophages resulted in a significant increase in infarct size and deterioration of left ventricular function after I/R injury, highlighting a beneficial effect of recruited macrophages in cardiac healing.

## Discussion

Macrophages are key players in cardiac homeostasis and disease (*Bajpai et al., 2019*; *Bajpai et al., 2018*; *Dick et al., 2019*; *Epelman et al., 2014*; *Hulsmans et al., 2017*; *Nahrendorf et al., 2007*; *Nicolás-Ávila et al., 2020*; *Panizzi et al., 2010*; *Sager et al., 2016*). The precise understanding of their developmental origin, their functions, and their regulation could enable the identification of macrophage-targeted strategies to modify inflammation in the heart. In cardiac repair after myocardial infarction, macrophages have both positive and negative effects. They are critical for tissue repair, angiogenesis, and inflammation regulation, but their actions need to be carefully balanced to prevent excessive inflammation, scar tissue formation, and adverse remodeling. This study sheds light on the differential role of resident and recruited macrophages in cardiac remodeling and outcome after AMI.

BM-derived recruited macrophages represent a small population in the healthy heart but are recruited in vast numbers to the injured myocardium after I/R injury. These recruited cells exhibit substantially different transcriptional profiles in comparison to their resident counterpart, and show proinflammatory properties. Resident macrophages remain present in the remote and border zones and display a reparative gene expression profile after I/R injury. In comparison to recruited macrophages, resident macrophages expressed higher levels of genes related to homeostatic functions (e.g. *Lyve1*, *Timd4*, *Cd163*, *Stab1*). These markers have recently been associated with self-renewing tissue macrophages that are maintained independently of BM contribution (*Dick et al., 2022*). Biological processes associated with cardiac healing are upregulated in resident macrophages (e.g. regulation of vascular development, regulation of cardiomyocyte development, extracellular matrix organization). Transcriptional profiles of recruited macrophages show enhanced inflammatory biological processes also in macrophages harvested from the remote area, underlining their potentially detrimental influence on remote cardiac injury.

Two recent studies addressed the role of resident macrophages using DT-mediated macrophage depletion and reported impaired cardiac remodeling in chronic myocardial infarction (*Dick et al., 2019*) and after I/R injury (*Bajpai et al., 2019*). However, DT-mediated cell ablation is known to induce neutrophil recruitment and tissue inflammation (*Frieler et al., 2015*; *Oh et al., 2017*; *Ruedl and Jung, 2018*; *Sivakumaran et al., 2016*). This inflammatory preconditioning of cardiac tissue after DT-depletion is likely to impact on cardiac remodeling and influence assessment of tissue macrophage functions. Genomic deletion of FIRE in the *Csf1r* gene results in the near-absence of resident cardiac macrophages but circumvents the inflammatory stimulus of DT-induced depletion. ΔFIRE specifically diminished homeostatic and antigen-presenting macrophages. Substantiating the specificity to resident cardiac macrophages in this genetic model, the amount of macrophages recruited to the infarct area after I/R injury was not affected in ΔFIRE mice. Furthermore, scRNA-seq analysis revealed that gene expression of monocytes as well as *Ccr2^hiLy6c^hi* and *Cx3cr1^hi* Mφ, representing mainly recruited immune cells, was mostly unchanged in ΔFIRE compared to control mice. The specific targeting of the resident macrophage population introduces an interesting in vivo model for studying their role in MI without the need for inflammation-prone conditional deletion of macrophages (*Frieler et al., 2015*).

Ablation of resident macrophages altered macrophage crosstalk to non-macrophage immune cells, especially lymphocytes and neutrophils. This was characterized by a proinflammatory gene signature, such as neutrophil expression of inflammasome-related genes and a reduction in anti-inflammatory genes like *Chil3* and *Lcn2* (*Gordon and Martinez, 2010*; *Guo et al., 2014*; *Parmar et al., 2018*). Interestingly, inflammatory polarization of neutrophils have also been associated with poor outcome

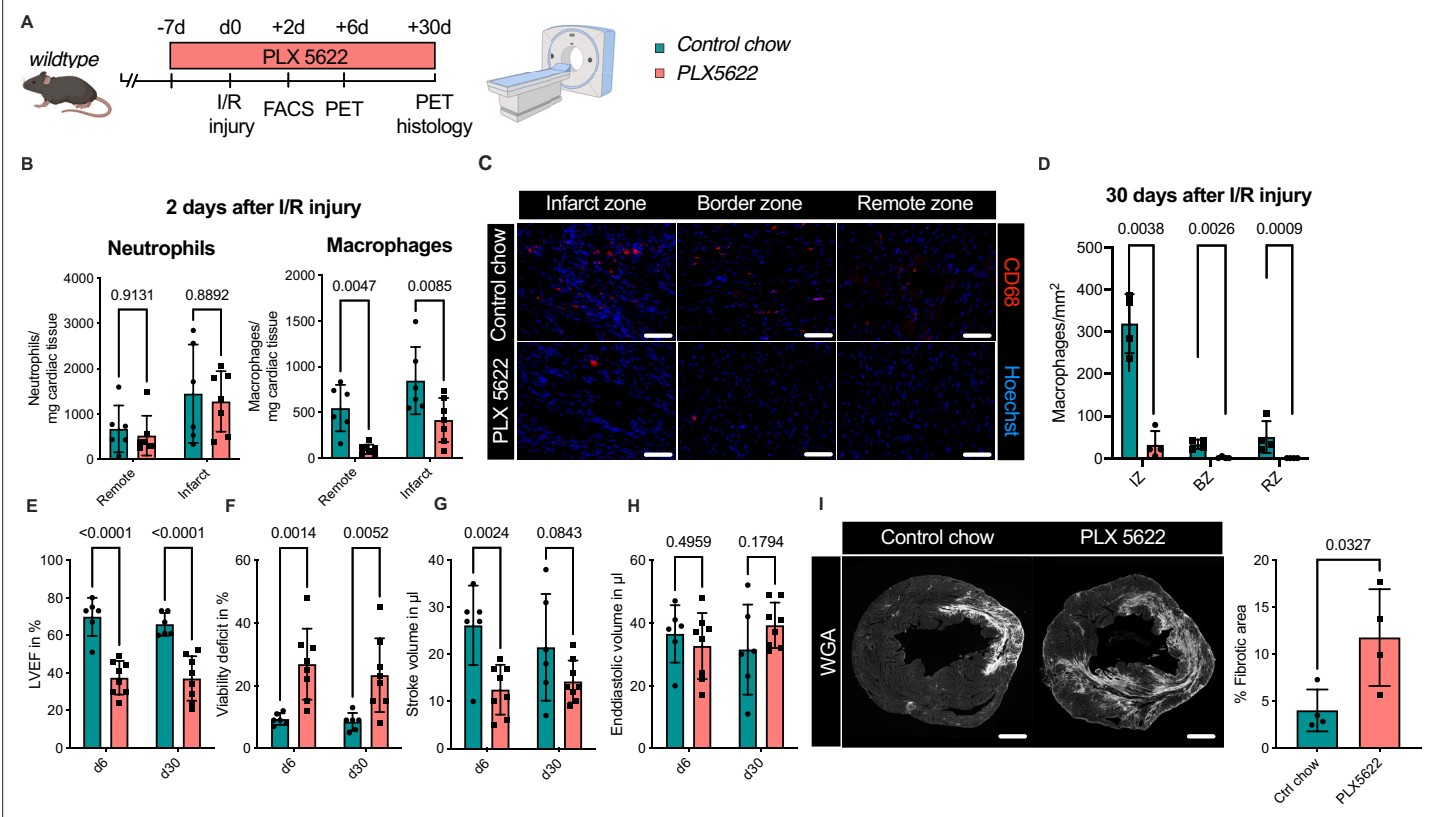

**Figure 8.** Ablation of resident and recruited macrophages severely impacts on cardiac healing after ischemia/reperfusion (I/R) injury. (**A**) Schematic of the analysis of cardiac function and infarct size in mice treated with PLX5622 7 days prior and 30 days after I/R injury. (**B**) Number of cardiac macrophages and neutrophils in the remote and ischemic myocardium 2 days after I/R injury in mice fed control chow (n = 6) or PLX5622 (n = 7). (**C**) Representative immunohistology of hearts 30 days after I/R injury showing macrophages (CD68+ cells in red and Hoechst in blue) in the infarct, border, and remote zones and (**D**) number of cardiac macrophages in the respective area (n = 4 for control chow and n = 4 for PLX5622). (**E**) Left ventricular ejection fraction (LVEF), (**F**) viability deficit, (**G**) stroke volume, and (**H**) end-diastolic volume measured using positron emission tomography 6 and 30 days after I/R injury (n = 6 for control chow, n = 8 for PLX5622). (**I**) Representative immunohistological images showing the fibrotic area (WGA+ area) in hearts 30 days after I/R injury from mice fed control chow or PLX5622. Percentage of fibrotic area in the respective groups (n = 4 for each group). Student's t-test or Fisher's Fisher's Least Significant Difference (LSD) test was performed and mean ± standard deviation (SD) is shown.

The online version of this article includes the following figure supplement(s) for figure 8:

**Figure supplement 1.** Macrophage depletion using the Csf1r-inhibitor PLX5622.

after ischemic brain injury (*Cuartero et al., 2013*). Clinical trials in myocardial infarction patients show a correlation of inflammatory markers with extent of the myocardial damage (*Sánchez et al., 2006*) and with short- and long-term mortality (*Mueller et al., 2002*).

Our study provides evidence that the absence of resident macrophages negatively influences cardiac remodeling in the late postinfarction phase in ΔFIRE mice indicating their biological role in myocardial healing. In the early phase after I/R injury, absence of resident macrophages had no significant effect on infarct size or LV function. These observations potentially indicate a protective role in the chronic phase after myocardial infarction by modulating the inflammatory response, including adjacent immune cells like neutrophils or lymphocytes.

Deciphering in detail the specific functions of resident macrophages is of considerable interest but requires both cell-specific and temporally controlled depletion of respective immune cells in injury, which to our knowledge is not available at present. These experiments could be important to tailor immune-targeted treatments of myocardial inflammation and postinfarct remodeling.

Depletion of macrophages by pharmacological inhibition of CSF1R induced the absence of both resident and recruited macrophages, allowing us to determine cardiac outcome in juxtaposition to the ΔFIRE mice. Continuous CSF1R inhibition induced the absence of macrophages also in the infarcted area and had deleterious effect on infarct size and LV function. In line with our findings, depletion of

macrophages by anti-CSF1R treatment was associated with worsened cardiac function in a model of pressure overload induced heart failure (*Revelo et al., 2021*). Controversially, a recent study in which monocytes were depleted using DT-injections in *Ccr2*^DTR mice reported beneficial effects on cardiac outcome after MI in mice (*Bajpai et al., 2019*). An explanation of this controversy might be the timing and duration of macrophage depletion. *Bajpai et al., 2019* depleted recruited macrophages only in the initial phase of myocardial infarction which improved cardiac healing, while depletion of macrophages over a longer period of time, as shown in our study, is detrimental for cardiac repair. Furthermore, other immune cells including neutrophils express CCR2 and may therefore be affected directly (*Talbot et al., 2015*; *Xu et al., 2017*). In line with this, Ccr2-deficient mice exhibit reduced acute recruitment of neutrophils to the brain after I/R injury, which was associated with reduced infarct size and brain edema (*Dimitrijevic et al., 2007*).

Taken together our study underlines the heterogeneity of cardiac macrophages and the importance of ontogeny therein. Resident macrophages, which mainly derive from YS EMPs, govern cardiac homeostasis in the healthy heart, and contribute positively to cardiac healing after I/R injury by orchestrating anti-inflammatory programming of other cardiac immune cells. However, recruited macrophages contribute to the healing phase after AMI and their absence defines infarct size and cardiac outcome.

## Materials and methods

### Mice

*Tnfsf11a*^Cre (*Jacome-Galarza et al., 2019*; *Percin et al., 2018*), *Flt3*^Cre (*Benz et al., 2008*), *Myb*^fl/fl (*Emambokus et al., 2003*), *Mx1*^Cre (*Kühn et al., 1995*) (from The Jackson Laboratory (JAX), Stock No: 003556), *Csf1r*^ΔFIRE (*Rojo et al., 2019*), *Rosa26*^fs-DTR (*Buch et al., 2005*) (JAX Stock No: 007900), *Rosa26*^fs-mT/mG (*Muzumdar et al., 2007*) (JAX Stock No: 007676), *Rosa26*^fs-eYFP (*Srinivas et al., 2001*) (JAX Stock No: 006148), and *Rosa26*^fs-RFP (JAX Stock No: 034720) mice have been previously described. PCR genotyping was performed according to protocols described previously. Animals were aged between 10 and 16 weeks. We have complied with all relevant ethical regulations. Animal studies were approved by the local regulatory agency (Regierung von Oberbayern, Munich, Germany, record numbers ROB-55.2-2532.Vet_02-19-17 and ROB-55.2-2532.Vet_02-19-1; Care and Use Committee of the Institut Pasteur (CETEA), dap190119).

FIRE mice were kept on a CBA/Ca background. Experiments in which reporter mice were necessary (*Csf1r*^ΔFIRE*Tnfsf11a*^Cre*Rosa26*^fs-eYFP) the background was mixed with a C57Bl6 background. All experiments evaluating cardiac function and outcome after infarction in FIRE mice were performed on mice kept with a CBA/Ca background.

For fate-mapping analysis of Flt3^+ precursors, *Flt3*^Cre males (the transgene is located on the Y chromosome) were crossed to *Rosa26*^fs-eYFP or *Rosa26*^fs-mT/mG female reporter mice and only the male progeny was used. For all other experiments, the offspring of both sexes was used for experiments.

BM transplantation was enabled by conditional deletion of the transcription factor *myb* as previously described (*Stremmel et al., 2018*). In brief, we induced BM ablation by four injections of polyI:C into *CD45.2; Mx1*^Cre*Myb*^flox/flox every other day, and then transplanted BM from congenic *C57BL/6 CD45.1* (*Ly5.1*; JAX Stock No: 006584). We confirmed the success (chimerism of above 90%) of the BM transplantation after 4 weeks.

To deplete macrophages, we used the selective CSF1R-inhibitor PLX5622. Control and PLX5622 (300 ppm formulated in AIN-76A standard chow, Research Diets, Inc) chows were kindly provided by Plexxikon Inc (Berkeley, CA). For depletion of embryonic macrophages, *Tnfsf11a*^Cre*Rosa26*^fs-DTR mice were injected intraperitoneally with 0.02 mg/kg body weight DT (Sigma-Aldrich) at the time points mentioned in the figure and corresponding legend. Mice were closely monitored by veterinarians and according to score sheets that were approved by the regulatory agency. Single DT application in *Tnfsf11a*^Cre*Rosa26*^fs-DTR mice let to rapid deterioration of their health (without infarction or other interventions), and consequently experiments needed to be aborted according to regulations; experimental outcomes were then documented as premature death as indicated.

### Ischemia–reperfusion (I/R) injury

I/R injury was carried out as previously described (*Novotny et al., 2018*). In brief, mice were anesthetized using 2% isoflurane and intraperitoneal injection of fentanyl (0.05 mg/kg), midazolam (5.0 mg/

kg), and medetomidine (0.5 mg/kg), and then intubated orally (MiniVent Ventilator model nr. 845, Harvard Apparatus) and ventilated (volume of 150 µl at 200 /min). After lateral thoracotomy, the left anterior descending artery (LAD) was ligated with an 8–0 prolene suture producing an ischemic area in the apical LV. To induce the reperfusion injury, the suture was removed after 60 min and reperfusion was confirmed by observing the recoloring of the LV. Postoperative analgesia was performed by injection of Buprenorphin (0.1 mg/kg) twice per day for 3 days. After 2, 6, or 30 days after I/R injury organs were harvested after cervical dislocation.

## Organ harvest

Mice were anesthetized using 2% isoflurane and organ harvest was performed after cervical dislocation. Blood was harvested with a heparinized syringe (2 ml) by cardiac puncture. After perfusion with 20 ml of ice-cold phosphate-buffered saline (PBS) hearts were excised and kept in PBS on ice until further tissue processing. For flow cytometry, hearts were divided into the ischemia (tissue distal of the LAD ligation) and remote area (tissue proximal of the LAD ligation). For histological examinations of tissues, hearts were incubated in 4% paraformaldehyde for 30 min followed by an incubation in 30% sucrose solution (Sigma-Aldrich) for 24 hr. Afterwards, hearts were mounted onto a heart slicing device and cut transversally into three equal parts (termed levels 1, 2, and 3) and stored in Tissue-Tek (Sakura Finetek Germany GmbH) at −80°C.

## Immunohistology

Cryosections (10–12 µm) of heart tissue were fixed with 4% paraformaldehyde for 10 min. Blocking and permeabilization were performed with 0.5% Saponin and 10% goat serum for 1 hr. Primary antibodies were added and incubated for 2–18 hr (see Table S1 - *Supplementary file 1*). Slides were washed with PBS and secondary antibodies were added and incubated for 1 hr (see Table S1 - *Supplementary file 1*). WGA-staining (Wheat Germ Agglutinin Alexa Fluor 647 conjugated antibody, Thermo Fisher Scientific) was used to locate and measure the infarct area and nuclei were stained with 4',6-diamidino-2-phenylindole (DAPI). Finally, slides were washed one more time and Fluorescence Mounting Medium was used to cover the stained sample.

Heart samples were evaluated using an Axio Imager M2 (Carl Zeiss) and blinded picture analysis was performed using ZEN Imaging and Axiovision SE64 Rel. 4.9.1 (Carl Zeiss). For the evaluation of cell numbers, six individual high-resolution images from each respective anatomical region (infarct area, border zone, and remote zone) were analyzed for each animal. To measure infarct size the heart was cut into three parts and the infarcted area was measured as WGA+ area in sections from each part.

## Flow cytometry

100 µl of heparinized blood was used for FACS analysis. Erythrocytes were lysed with 1% ammonium chloride. After washing with PBS, the cell suspension was resuspended in purified rat anti-mouse CD16/CD32 (BD Pharmingen) and incubated for 15 min at 4°C. Following this, cells were incubated with FACS antibodies for 15 min at 4°C (*Supplementary file 1*).

Heart tissue was dissected into remote and ischemic tissue as described above and minced into small pieces using forceps and a scalpel. When comparing baseline and I/R injury in FACS analysis, basal heart tissue was used for comparison with remote tissue and apical heart tissue for comparison with ischemia tissue. After enzymatic digestion (Collagenase XI 1200 U/mg, Collagenase I 125 U/mg, Hyaluronidase 500 U/mg, DNase I 1836 U/mg; Sigma-Aldrich) for 30 min at 37°C cells were washed and incubated with purified anti-CD16/32 (FcγRIII/II; dilution 1/50) for 10 min. Thereafter, cells were incubated with FACS antibodies (*Supplementary file 1*) for 30 min at 4°C. FACS analysis was performed on a BD Fortessa or a BeckmanCoulter Cytoflex flow cytometer and gating strategies are shown in *Figure 2—figure supplement 1*. Data were analyzed using FlowJo (version 10.0.8r1).

## Cell sorting

Cell sorting was performed on a MoFlo Astrios (Beckman Coulter) to obtain cardiac macrophages from *CD45.2; Mx1*$^{Cre}$*Myb*$^{flox/flox}$ after BM transplantation of CD45.1 BM (*n* = 3 for 2 days after I/R injury) for bulk sequencing, or all cardiac immune cells (CD45+ cells) of *Csf1r*$^{ΔFIRE}$ and *Tnfsf11a*$^{Cre}$*Rosa26*$^{fs-eYFP}$ mice (*n* = 3 for *Csf1r*$^{ΔFIRE/+}$ and *Csf1r*$^{ΔFIRE/ΔFIRE}$ in baseline conditions, *n* = 1 for *Tnfsf11a*$^{Cre}$*Rosa26*$^{fs-eYFP}$ in baseline conditions; n=2 for *Csf1r*$^{ΔFIRE/+}$ and *Csf1r*$^{ΔFIRE/ΔFIRE}$ 2 days after I/R injury) for single-cell analysis.

CD45$^+$ cells were enriched by using magnetic beads and MS columns (CD45 MicroBeads; Miltenyi Biotec). Cells were then sorted as single/live/CD45$^+$ cells for single-cell sequencing from baseline hearts and from the ischemic myocardium 2 days after I/R injury. Bulk sequencing was performed on single/live/CD45$^+$/lin$^-$/CD11b$^+$/F4/80$^+$/CD64$^+$/CD45.1$^+$ or CD45.2$^+$ cells from the remote and the ischemic myocardium 2 days after I/R injury. Dead cells were identified with SYTOX Orange Dead Cell Stain.

## Bulk sequencing and analysis

For each sample, ~1000 macrophages were sorted into 75 µl of RLT buffer (QIAGEN, containing 1% beta-mercaptoethanol), vortexed for 1 min and immediately frozen (−80°C). RNA extraction (RNeasy Plus Micro Kit, QIAGEN), cDNA generation (SMART-Seq v4 Ultra Low Input RNA Kit, Takara Bio), and library preparation (Nextera XT DNA Library Prep Kit, Illumina) were performed according to the manufacturer's specifications. Sequencing was performed on a HiSeq4000 system (Ilumina).

The obtained reads were trimmed using bbduk from the BBMap (https://sourceforge.net/projects/bbmap/) v38.87 collection using parameters "ktrim = r k=23 mink = 11 hdist = 1 tpe tbo". The trimmed reads were aligned with Hisat2 2.2.1 against the Ensembl release 102 reference mouse genome (*Yates et al., 2020*).

Gene expression was quantified using the featureCounts (*Liao et al., 2014*) application from the subread package (v2.0.1) and with parameters '--primary -O -C -B -p -T 8 --minOverlap 5'. Differential expression analysis was performed using DESeq2 (v1.30.0) (*Love et al., 2014*). A gene ontology set enrichment analysis was performed using the ClusterProfiler R package (v3.18.1). The visualization of the respective clustermaps was layouted using the ForceAtlas2 implementation of the python fa2 (https://pypi.org/project/fa2/) package for networkx (v2.5) (https://networkx.org/).

On the library-size normalized count data, the pymRMR (*Peng et al., 2005*, https://pypi.org/project/pymrmr/) package was used to derive the top 100 discriminatory genes (Mutual Information Quotient method) for subsequently calculating the UMAP 2D-embedding (*McInnes et al., 2020*, https://pypi.org/project/umap-learn/) for all samples (umap-learn package v0.5.0rc1, 3 neighbors) (*McInnes et al., 2018*, https://pypi.org/project/pymrmr/).

## Single-cell RNA sequencing and analysis

After sorting, cells were proceeded for single-cell capture, barcoding and library preparation using Chromium Next GEM single cell 3' (v3.1, 10× Genomics) according to the manufacturer's specifications. Pooled libraries were sequenced on an Illumina HiSeq1500 sequencer (Illumina, San Diego, USA) in paired-end mode with asymmetric read length of 28 + 91 bp and a single indexing read of 8 bp.

The reads of heart1 sample (*Tnfsf11a$^{Cre}$Rosa26$^{fs-eYFP}$*) were demultiplexed using Je-demultiplex-illu (*Girardot et al., 2016*) and mapped against a customized mouse reference genome (GRCm38.p6, Gencode annotation M24) including eYfp sequence using CellRanger (v3.1.0, 10× Genomics).

The six mouse samples 20133-0001 to 20133-0006 (baseline condition of *Csf1r$^{ΔFIRE/+}$* and *Csf1r$^{ΔFIRE/ΔFIRE}$*), were processed using Cellranger 4.0.0 using the 2020 A mm10 reference.

The four mouse samples (MUC13956-13959, infarct condition of *Csf1r$^{ΔFIRE/+}$* and *Csf1r$^{ΔFIRE/ΔFIRE}$*) were sequenced with four technical sequencing replicates and pooled using the cellranger (v4.0.0, 10× Genomics) pipeline. Cellranger 4.0.0 was called with default parameters and the 2020 A mm10 reference for gene expression.

Finally, all 11 samples were integrated using Seurat 4.0.0 (on R 4.0.1) (*Stuart and Satija, 2019*). The samples were processed, and cells were filtered to contain between 200 and 6000 features, have at least 1000 molecules detected (nCount_RNA >1000), have below 15% mtRNA content (^MT) and below 40% ribosomal RNA content (^Rps|^Rpl). After this filtering a total of 35,759 cells remained.

After performing SCTransform (*Hafemeister and Satija, 2019*) on the samples, the SCTransform vignette for integrating the datasets was followed (with 2000 integration features). For dimensionality reduction, PCA was performed using default parameters, and UMAP and Neighbour-Finding was run on 50 PCs. Clustering was performed at a resolution of 0.8. A total of 18 clusters was identified using this approach. Cluster markers were calculated using the *t*-test in the FindMarkers function. Subsequently, cell types were initially predicted using the cPred cell-type prediction (https://github.com/mjoppich/scrnaseq_celltype_prediction, copy archived at *Joppich, 2023a*). Upon manual curation,

further fine-grained cell-type annotations were made. Differential comparisons were performed against several subgroups of the dataset. These comparisons were performed using the *t*-test in the FindMarkers function of Seurat. Differential results are visualized using the EnhancedVolcano library (*Blighe and Lewis, 2021*).

For the analysis of cell–cell interactions we downloaded the LR pairs from *Jin et al., 2021* from the Lewis Lab GitHub repository (https://github.com/LewisLabUCSD/Ligand-Receptor-Pairs, copy archived at *Armingol, 2024*). For each interaction (LR pair for a cluster pair) the communication score is calculated as the expression product *Armingol et al., 2021* of the mean normalized expressions exported from the Seurat object. This ensures that little expression of either ligand or receptor in only one cluster results in a relatively low communication score, and only good expression of ligand and receptor will result in a high communication score. The direction of an interaction is fixed from ligand to receptor. The single LR communication scores were then aggregated (sum) such that only interactions with a score greater 6 were taken into account.

Gene module scores for inflammasome, reactive oxygen species (ROS), and phagocytosis gene sets were calculated using Seurat's AddModuleScore function. All scripts, including the ones for creating the visualizations of bulk and scRNA-seq data, are available online through https://github.com/mjoppich/myocardial_infarction, copy archived at *Joppich, 2023b*.

## In vivo PET imaging

Electrocardiogram (ECG)-gated PET images were performed on days 6 and 30 after I/R injury of the LAD using a dedicated small-animal PET scanner (Inveon Dedicated PET, Preclinical Solutions, Siemens Healthcare Molecular Imaging, Knoxville, TN, USA), as previously described (*Brunner et al., 2012*). Anaesthesia was induced with isoflurane (2.5%), delivered via a face mask in pure oxygen at a rate of 1.2 l/min and maintained with isoflurane (1.5%). Approximately 15 MBq 2-deoxy-2-[$^{18}$F]fluoro-D-glucose ([$^{18}$F]-FDG) (~100 µl) were administered through a teil vein catheter and slowly flushed immediately afterwards with 50 µl saline solution. Body temperature was monitored using a rectal thermometer and maintained within the normal range using a heating pad. After placing animals within the aperture of the PET scanner ECG electrodes (3 M, St. Paul, MN, USA) were placed on both forepaws and the left hind paw and ECG was recorded using a dedicated physiological monitoring system (BioVet; Spin Systems Pty Ltd, Brisbane, Australia) (*Todica et al., 2018*). The PET emission acquisition (list-mode) was initiated 30 min after [$^{18}$F]-FDG injection and lasted 15 min (*Brunner et al., 2012*; *Gross et al., 2016*). For scatter and attenuation correction and additional 7-min long transmission scan was performed using a Co-57 source.

The accuracy of the ECG trigger signal was verified retrospectively using in-house software programmed in MATLAB (The Mathworks, Natick, USA) and in C programming language and erroneous trigger events were removed when needed, as previously described (*Böning et al., 2013*). Further processing of the data was performed using the Inveon Acquisition Workplace (Siemens Medical Solutions, Knoxville, TN). As previously described, data were reconstructed as a static image or as a cardiac gated image with 16 bins in a 128 × 128 matrix with a zoom of 211% using an OSEM 3D algorithm with 4 and a MAP 3D algorithm with 32 iterations (*Brunner et al., 2012*). The reconstructed data were normalized, corrected for randoms, dead time, and decay as well as attenuation and scatter.

PET images were analyzed using the Inveon Research Workplace in a blinded manner (Siemens Medical Solutions, Knoxville, TN). Infarct sizes were determined from static reconstructed images using QPS (Cedars-Sinai, Los Angeles, CA, USA). Hereby, datasets were compared to a normative database and the viability defect was calculated as percentage of the left ventricular volume, as described previously (*Lehner et al., 2014a*; *Lehner et al., 2014b*). Left ventricular function volumes (end-diastolic volume (EDV), end-systolic volume (ESV), and stroke volume (SV)), as well as the LVEF, were determined from ECG-gated images using QGS (Cedars-Sinai, Los Angeles, CA, USA), as described previously (*Brunner et al., 2012*; *Croteau et al., 2003*).

## Statistical analysis

Student's *t*-test or Fisher's Fisher's Least Significant Difference (LSD) test was used (Prism GraphPad). Welch's correction for unequal variances was used when applicable. A p-value of <0.05 was considered

significant. The Shapiro–Wilk test was used to test normality. Data are presented as mean ± standard deviation.

## Acknowledgements

We thank Nicole Blount, Beate Jantz, Michael Lorenz, and Sebastian Helmer for excellent technical assistance. DZHK (German Centre for Cardiovascular Research) and the BMBF (German Ministry of Education and Research) (grants 81Z0600204 to CS, 81X2600252 to TW and 81X2600256 to MF). The Deutsche Forschungsgemeinschaft SCHU 2297/1-1 and collaborative research centers 1123 project Z02 (MJ, RZ) and A07 (CS), and CRC TRR332 project A6 (CS). LMUexcellent (TW). ESC Research Grant 2021 (TW). Friedrich-Baur Stiftung (TW). Deutsche Stiftung für Herzforschung (MF). Chinese Scholarship Council (CSC) to JF and LL.

## Additional information

### Funding

| Funder | Grant reference number | Author |
|---|---|---|
| Deutsches Zentrum für Herz-Kreislaufforschung | 81Z0600204 | Christian Schulz |
| Deutsches Zentrum für Herz-Kreislaufforschung | 81X2600252 | Tobias Weinberger |
| Deutsches Zentrum für Herz-Kreislaufforschung | 81X2600256 | Maximilian Fischer |
| Deutsche Forschungsgemeinschaft | SCHU 2297/1-1 | Christian Schulz |
| Deutsche Forschungsgemeinschaft | SFB 1123 project Z02 | Ralf Zimmer |
| Deutsche Forschungsgemeinschaft | SFB 1123 project A07 | Christian Schulz |
| Deutsche Forschungsgemeinschaft | SFB TRR332, project A06 | Christian Schulz |
| LMUexcellent | | Tobias Weinberger |
| European Society of Cardiology | Research Grant 2021 | Tobias Weinberger |
| Friedrich-Baur Stiftung | | Tobias Weinberger |
| Deutsche Stiftung für Herzforschung | | Maximilian Fischer |
| Chinese Scholarship Council | | Jiahui Fang Lulu Liu |

The funders had no role in study design, data collection, and interpretation, or the decision to submit the work for publication.

### Author contributions

Tobias Weinberger, Conceptualization, Formal analysis, Supervision, Funding acquisition, Validation, Visualization, Methodology, Writing – original draft, Writing – review and editing; Messerer Denise, Maximilian Fischer, Clarisabel Garcia Rodriguez, Saskia Räuber, Lukas Thomas, Investigation, Methodology, Writing – review and editing; Markus Joppich, Software, Formal analysis, Visualization, Methodology, Writing – review and editing; Konda Kumaraswami, Sonja Ablinger, Jiahui Fang, Lulu Liu, Wing Han Liu, Julia Winterhalter, Johannes Lichti, Dena Esfandyari, Guelce Percin, Investigation, Methodology; Vanessa Wimmler, Sandra Matin, Investigation; Andrés Hidalgo, Stefan Engelhardt, Conceptualization; Claudia Waskow, Conceptualization, Resources; Andrei Todica, Conceptualization, Investigation, Visualization, Methodology; Ralf Zimmer, Conceptualization, Data curation,

Investigation, Writing – review and editing; Clare Pridans, Conceptualization, Resources, Methodology; Elisa Gomez Perdiguero, Conceptualization, Methodology, Writing – review and editing; Christian Schulz, Conceptualization, Validation, Investigation, Visualization, Methodology, Writing – original draft, Project administration, Writing – review and editing

## Author ORCIDs
Tobias Weinberger  https://orcid.org/0000-0003-0024-020X
Markus Joppich  http://orcid.org/0000-0002-6665-8951
Maximilian Fischer  http://orcid.org/0000-0001-9172-3316
Saskia Räuber  http://orcid.org/0000-0001-8901-5572
Dena Esfandyari  http://orcid.org/0000-0001-8732-1461
Stefan Engelhardt  http://orcid.org/0000-0001-5378-8661
Clare Pridans  http://orcid.org/0000-0001-9423-557X
Christian Schulz  http://orcid.org/0000-0002-8149-0747

## Ethics
Animal studies were approved by the local regulatory agency (Regierung von Oberbayern, Munich, Germany, record numbers ROB-55.2-2532.Vet_02-19-17 and ROB-55.2-2532.Vet_02-19-1; Care and Use Committee of the Institut Pasteur (CETEA), dap190119).

Reviewer #1 (Public Review): https://doi.org/10.7554/eLife.89377.4.sa1
Reviewer #2 (Public Review): https://doi.org/10.7554/eLife.89377.4.sa2
Author response https://doi.org/10.7554/eLife.89377.4.sa3

# Additional files

## Supplementary files
• Supplementary file 1. Antibody list.

• MDAR checklist

## Data availability
Sequencing data have been deposited in GEO under accession codes GSE263544, GSE263545.

The following datasets were generated:

| Author(s) | Year | Dataset title | Dataset URL | Database and Identifier |
|---|---|---|---|---|
| Joppich M, Weinberger T, Schulz C | 2024 | Resident and recruited macrophages differentially contribute to cardiac healing after myocardial ischemia [bulk RNA-seq] | https://www.ncbi.nlm.nih.gov/geo/query/acc.cgi?acc=GSE263544 | NCBI Gene Expression Omnibus, GSE263544 |
| Joppich M, Weinberger T, Schulz C | 2024 | Resident and recruited macrophages differentially contribute to cardiac healing after myocardial ischemia [scRNA-seq] | https://www.ncbi.nlm.nih.gov/geo/query/acc.cgi?acc=GSE263545 | NCBI Gene Expression Omnibus, GSE263545 |

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
