## [Editor Report · eLife assessment]

Using state-of-the-art fate-mapping models and genetic and pharmacological targeting approaches, this study provides **important** findings on the distinct functions exerted by resident and recruited macrophages during cardiac healing after myocardial ischemia. Evidence supporting the conclusions are **solid** with the use of the FIRE mouse model in combination with fate-mapping to target fetal-derived macrophages. This study will be of interest for the macrophage biologists working in the heart but also in others tissues in the context of inflammation.

---

## [Referee Report · Reviewer #1 (Public Review)]

Weinberger et al. use different fate-mapping models, the FIRE model and PLX-diet to follow and target different macrophage populations and combine them with single-cell data to understand their contribution to heart regeneration after I/R injury. This question has already been addressed by other groups in the field using different models. However, the major strength of this manuscript is the usage of the FIRE mouse model that, for the first time, allows specific targeting of only fetal-derived macrophages.

The data show that the absence of resident macrophages is not influencing infarct size but instead is altering the immune cell crosstalk in response to injury, which is in line with the current idea in the field that macrophages of different origins have distinct functions in tissues, especially after an injury.

To fully support the claims of the study, specific targeting of monocyte-derived macrophages or the inhibition of their influx at different stages after injury would be of high interest.

In summary, the study is well done and important for the field of cardiac injury. But it also provides a novel model (FIRE mice + RANK-Cre fate-mapping) for other tissues to study the function of fetal-derived macrophages while monocyte-derived macrophages remain intact.

---

## [Referee Report · Reviewer #2 (Public Review)]

In this study Weinberger et al. investigated cardiac macrophage subsets after ischemia/reperfusion (I/R) injury in mice. The authors studied a ∆FIRE mouse model (deletion of a regulatory element in the Csf1r locus), in which only tissue resident macrophages might be ablated. The authors showed a reduction of resident macrophages in ∆FIRE mice and characterized its macrophages populations via scRNAseq at baseline conditions and after I/R injury. 2 days after I/R protocol ∆FIRE mice showed an enhanced pro inflammatory phenotype in the RNAseq data and differential effects on echocardiographic function 6 and 30 days after I/R injury. Via flow cytometry and histology the authors confirmed existing evidence of increased bone marrow-derived macrophage infiltration to the heart, specifically to the ischemic myocardium. Macrophage population in ∆FIRE mice after I/R injury were only changed in the remote zone. Further RNAseq data on resident or recruited macrophages showed transcriptional differences between both cell types in terms of homeostasis-related genes and inflammation. Depleting all macrophage using a Csf1r inhibitor resulted in a reduced cardiac function and increased fibrosis.

Strengths:

(1) The authors utilized robust methodology encompassing state of the art immunological methods, different genetic mouse models and transcriptomics.

(2) The topic of this work is important given the emerging role of tissue resident macrophages in cardiac homeostasis and disease.

Comments on revised version:

The authors have responded to all questions. I have no further comments and congratulate the authors on their work.

---

## [Author Response]

The following is the authors’ response to the previous reviews.

**Recommendations for the authors:**

**Reviewer #1 (Recommendations For The Authors):**
The authors have addressed most of the points that were made. However, despite some things that may well be beyond the scope, I would like to insist on a few small points:Point 1: If the authors have conducted a gross analysis of cardiac morphology by histology already, they should include this data in the manuscript and comment with 1-2 sentences that "cardiac healing"..."is unlikely influenced by developmental defects".

We agree with the reviewer that this analysis is important. Therefore, we are currently conducting an in-depth analysis of the cardiac phenotype of different mouse strains lacking distinct subpopulations of cardiac macrophages in development and non-stimulated (baseline) conditions, including functional, metabolic and even electrophysiological aspects. These analysis will also include FIRE mice. While a gross analysis in this mouse strain did not show pathologic aspects, we look forward to the very detailed tissue characterization before publishing any data from a first basic analysis.

Point 7: There is still no legend in Figure 6: what is read? What is blue?

We added the respective legend in the figure.

Point 8: Please add the information on the background of mice used for the different FIRE mice into the methods part of the paper

We added the information in the Methods Part (lines 344-347).

**Reviewer #2 (Recommendations For The Authors):**
The authors have responded to all questions. I have no further comments and congratulate the authors on their work.

We thank the reviewer for their important questions and the constructive feedback.